# Does the Fetus Limit Antibiotic Treatment in Pregnant Patients with COVID-19?

**DOI:** 10.3390/antibiotics11020252

**Published:** 2022-02-16

**Authors:** Tito Ramírez-Lozada, María Concepción Loranca-García, Claudia Erika Fuentes-Venado, Carmen Rodríguez-Cerdeira, Esther Ocharan-Hernández, Marvin A. Soriano-Ursúa, Eunice D. Farfán-García, Edwin Chávez-Gutiérrez, Xóchitl Ramírez-Magaña, Maura Robledo-Cayetano, Marco A. Loza-Mejía, Ivonne Areli Garcia Santa-Olalla, Oscar Uriel Torres-Paez, Rodolfo Pinto-Almazán, Erick Martínez-Herrera

**Affiliations:** 1Servicio de Ginecología y Obstetricia, Hospital Regional de Alta Especialidad de Ixtapaluca, Ixtapaluca 56530, Mexico; titolozada@yahoo.com.mx (T.R.-L.); ramaxo@hotmail.com (X.R.-M.); 2Hospital General de Zona No. 53, Los Reyes, Instituto Mexicano del Seguro Social (IMSS), Carr Federal México-Puebla Km 17.5, Villa de la Paz, Rincón de los Reyes, Los Reyes Acaquilpan 56400, Mexico; lorancagarcia@yahoo.com.mx; 3Servicio de Medicina Física y Rehabilitación, Hospital General de Zona No. 197 IMSS, Texcoco 56108, Mexico; cefvenado@hotmail.com; 4Efficiency, Quality, and Costs in Health Services Research Group (EFISALUD), Galicia Sur Health Research Institute (IIS Galicia Sur), SERGAS-UVIGO, 36213 Vigo, Spain; carmencerdeira33@gmail.com; 5Dermatology Department, Hospital Vithas Ntra. Sra. de Fátima, 36206 Vigo, Spain; 6Campus Universitario, University of Vigo, 36310 Vigo, Spain; 7Sección de Estudios de Posgrado e Investigación, Escuela Superior de Medicina, Instituto Politécnico Nacional, Mexico City 11340, Mexico; estherocharan@hotmail.com (E.O.-H.); msoriano@ipn.mx (M.A.S.-U.); efarfang@ipn.mx (E.D.F.-G.); 8Doctorado en Ciencias en Biomedicina y Biotecnología Molecular, Escuela Nacional de Ciencias Biológicas, IPN, Mexico City 07738, Mexico; chz_edwin.bioexp@hotmail.com; 9Unidad de Investigación, Hospital Regional de Alta Especialidad de Ixtapaluca, Estado de Mexico 56530, Mexico; mrobledoc@hotmail.com (M.R.-C.); msptorreshraei@hotmail.com (O.U.T.-P.); 10Design, Isolation, and Synthesis of Bioactive Molecules Research Group, Chemical Sciences School, Universidad La Salle-México, Benjamín Franklin 45, Mexico City 06140, Mexico; marcoantonio.loza@lasalle.mx; 11Subdirección de Enfermería, Hospital Regional de Alta Especialidad de Ixtapaluca, Ixtapaluca 56530, Mexico; mspbony@hotmail.com; 12Non-Communicable Disease Research Group, Facultad Mexicana de Medicina, Universidad La Salle-México, Las Fuentes 17, Tlalpan Centro I, Tlalpan, Mexico City 14000, Mexico

**Keywords:** antibiotics, pregnancy, COVID-19, fetus, microbiota, placenta

## Abstract

During pregnancy, there is a state of immune tolerance that predisposes them to viral infection, causing maternal-fetal vulnerability to the adverse effects of COVID-19. Bacterial coinfections significantly increase the mortality rate for COVID-19. However, it is known that all drugs, including antibiotics, will enter the fetal circulation in a variable degree despite the role of the placenta as a protective barrier and can cause teratogenesis or other malformations depending on the timing of exposure to the drug. Also, it is important to consider the impact of the indiscriminate use of antibiotics during pregnancy can alter both the maternal and fetal-neonatal microbiota, generating future repercussions in both. In the present study, the literature for treating bacterial coinfections in pregnant women with COVID-19 is reviewed. In turn, we present the findings in 50 pregnant women hospitalized diagnosed with SARS-CoV-2 without previous treatment with antibiotics; moreover, a bacteriological culture of sample types was performed. Seven pregnant women had coinfection with *Staphylococcus haemolyticus*, *Staphylococcus epidermidis*, *Streptococcus agalactiae*, *Escherichia coli* ESBL +, biotype 1 and 2, *Acinetobacter jahnsonii*, *Enterococcus faecium*, and *Clostridium difficile*. When performing the antibiogram, resistance to multiple drugs was found, such as macrolides, aminoglycosides, sulfa, dihydrofolate reductase inhibitors, beta-lactams, etc. The purpose of this study was to generate more scientific evidence on the better use of antibiotics in these patients. Because of this, it is important to perform an antibiogram to prevent abuse of empirical antibiotic treatment with antibiotics in pregnant women diagnosed with SARS-CoV-2.

## 1. Introduction

Since the COVID-19 pandemic was declared, until December 1st, 2021, the World Health Organization has confirmed a total number of cases and deaths [1,2,3]. This worldwide disease is known to seriously impact the health system in low- and middle-income countries. As for pregnant women, there is a state of immune tolerance during pregnancy that predisposes them to viral infection [4], causing vulnerability in both patients and newborns to the adverse effects of COVID-19 [2,5]. Recent viral outbreaks such as influenza, Ebola, or the present COVID-19 pandemic have shown that pregnant women are exposed to a worse outcome than the general population. During pregnancy, the immune system is transformed because the maternal immune system is influenced by alloantigens expressed by the placenta and the fetus. Among the changes that have been identified, there is an increased complement activity, a lower B cell function, a decrement of plasmatic pro-inflammatory cytokines (such as IL-2 and interferon-gamma), and greater concentrations of anti-inflammatory cytokines such as IL-4 and IL-10 have been reported [6]. 

According to the most recent information, it seems there is no higher risk of catching COVID-19 in pregnant women. As for SARS-CoV-2-positive pregnant women, reports indicate that about 80–90% are asymptomatic, and only 13–15% present mild symptomatology (fever or other symptoms). Apparently, pregnant women are also unlikely to experience serious illness, with only 5–8% who get severe disease and 1–3% becoming critical [7,8]

However, reports indicate that when there is a condition such as a viral infection that puts at risk this security, preterm labor, or even fetal death are a possibility [9,10]. Actually, pregnant women with COVID-19 had an increased risk of preeclampsia, preterm birth, and other adverse effects during pregnancy [11,12].

A possible explanation for the higher risk of pregnancy complications in infected SARS-CoV-2 women can be associated with the increase in the expression of ACE2 during gestation [13,14]. ACE is the central part of the renin–angiotensin system (RAS) and a crucial mechanism of blood pressure in mammals. ACE2 regulates vascular tension modulated by RAS because it can decompose ANG I to produce ANG 1–9, its most pleiotropic element, and may lyse ANG II to ANG 1–7 (vasodilatory bioactive peptide) [15,16,17]. Moreover, SARS-CoV-2 uses ACE2 to penetrate host cells [18]. Clinical and in vivo studies reported a significantly greater expression of ANG 1–7 and ACE2 in the kidney, placenta, and uterus during pregnancy compared to non-pregnant species [13,14]. Nevertheless, in pregnancy, the association between ACE2 upregulation and SARS-CoV-2 needs further study [19].

The recommendations for antibiotics use in COVID-19 infection were made based upon the in vitro activity against some viruses such as influenza AH1N1 and Zika, by Azithromycin or Doxycillin, regarding their in vitro activity with influenza virus and dengue [20].

## 2. Physiological Changes Related to the Medication Administration

The variability in response to administering a drug is a characteristic of all the treatments administered to healthy people. In the case of pregnant women, physiological changes amplify the variability of response to medications. Within them, the availability of a drug can be altered by those events that occur during pregnancy (Table 1), such as hormone level changes, fluid and fat volume, cardiac output, glomerular filtration rate, protein concentration, and drug-metabolizing enzymes. Each of them has the potential to affect pharmacokinetics [1,21,22,23,24,25].

There is evidence that pharmacokinetics is altered during pregnancy affecting drug distribution and metabolism. For example, CYP1A2 activity is decreased during all pregnancies, while CYP2B6 and CYP3A4 levels are increased in the last trimester [26]. Since these enzymes play an essential role in drug metabolism and elimination, including most antibiotics, the levels of drugs and their metabolites would be altered, which could lead to unexpected reactions, including treatment ineffectiveness or drug toxicity [27].

## 3. Passage through the Placenta

Practically all medications, including antibiotics, will enter the fetal circulation in a variable quantity regardless of the role of the placenta as a protective barrier, particularly those that are small (<500 Da) and nonpolar. This phenomenon happens primarily by passive diffusion and a smaller quantity travels by active transport and facilitates diffusion. The drugs’ physicochemical properties such as molecular weight, liposolubility, and polarity are crucial to the rate of passive transmission. Other essential elements of placental transfer involve primarily maternal–fetal blood flow, which fetal or maternal conditions can modify, and also by membrane thickness and exchange surface region [24,28].

The surface area of the chorionic villi and placental blood flow increases as gestation progress, implying that the transfer rate likewise rises with gestational age. Once the drugs, driven across by active transport to the placenta, reach the apical border of the syncytiotrophoblast, different transporter families such as human equilibrium nucleoside transporter (hENT1/2), novel organic cation transporters (OCTN), organic anion transport protein (OATP), and monocarboxylate transporters (MCT) permit the MRP (multi-resistance protein) and MDR (multi-resistance) transporters transcellular passage into fetal blood [24,28].

It is worth mentioning that the presence of these transporters fluctuates during pregnancy; hence, what can modulate the concentration of drugs that cross the placenta may vary in different stages of gestational age. Furthermore, in vitro studies have reported that inflammation also affects the expression of transporters. Finally, drugs structurally related to endogenous compounds can also be transported by facilitated diffusion [24,28].

## 4. The Risks of Using Antibiotics during the First Trimester

Congenital disabilities are present in 2–3% of newborns, and approximately 1–2% of this total is associated with exposure to teratogens. Teratogenesis is defined as dysgenesis of the fetal organs regarding structural integrity or function. It can be expressed as malformations that occur during organogenesis or at a later stage by altering the structure or operation of the apparatus or systems involved [29].

The fetal response is affected by several factors, including genetic predisposition, the dose of the teratogenic agent, pathway, and timing of exposure to the drug which is a critical factor in determining the nature and extent of any adverse effects [29].

The three crucial phases of human development are:
Pre-embryonic phase: Extends from conception to 17 days after conception. During this period, any adverse effect is an “all or nothing phenomenon,” and the result of an insult will be embryonic death or intact survival through the multiplication of totipotent cells.Embryonic phase: Comprises from day 18 to day 55. It is the period of greatest vulnerability for the embryo due to the rapid differentiation of tissues.Fetal phase (from the eighth week of gestation to term): The cerebral cortex and renal glomeruli continue to develop and remain susceptible to damage. Functional abnormalities such as deafness may occur, and drugs that can cross the placenta can affect fetal growth and development rather than causing structural malformations [29].

Reports indicate that approximately 80% of prescriptions during pregnancy are antibiotics, and it is assumed that between 20–39% of women will receive antibiotics during pregnancy [30,31,32]. During pregnancy, different antibacterial regimens are used in clinical practice, including cephalosporins, ẞ-lactams, macrolides, fluoroquinolones, glycopeptides, and lincosamides [30,31,32]. However, its ability to cross the placenta and reach the fetal compartments is variable, derived from conditions that affect the placenta, such as maternal diseases and gestational age (Figure 1).

## 5. Causal Agents

Coinfection pertains to simultaneous infection by two or more pathogens in a cell or host. On the other hand, superinfection is when a pathogen infects the host shortly before a second infection with another pathogen. In both cases, the state of the host depends mainly on the balance between the immunopathology exerted by the pathogens and the immunity of the host [33].

Coinfections and superinfections are usual in numerous viral infections associated with the respiratory system. Bacterial coinfections significantly increase the mortality rate of viral infected patients. The coinfection of SARS-CoV-2 with other microorganisms is a significant factor in the pathogenesis of COVID-19 that can complicate the precise diagnosis, treatment, and prognosis of the disease and even increase mortality rates. Clinical trials and metagenomic investigations indicated the co-presence of other viruses, bacteria, archaea, and fungi with SARS-CoV-2 in COVID-19 patients. Approximately 50% of the patients who died from SARS-CoV-2 had secondary bacterial infections that promoted COVID-19 pathophysiological progression [33].

In severe forms of SARS-CoV-2, patients showed higher biomarkers related to infection and cytokines, suggesting possible bacterial coinfection as a consequence of immune system dysregulation. Furthermore, antibiotic resistance can cause additional complications since coinfection with coronavirus and pneumonia could exceed the capacities of health care systems [34].

Recognition of SARS-CoV-2 infection is important as it allows the implementation of appropriate infection control measures and possible promising antiviral therapy. However, it should not ignore the possibility of coinfection. The discovery of new therapeutic agents should consider the mechanisms related to the synergy between COVID-19 and bacterial infections [33,34]. In bacterial and fungus coinfection studies, some co-pathogens such as *M. pneumoniae*, *Legionella pneumophila*, *Streptococcus pneumoniae*, *C. pneumoniae*, *Staphylococcus aureus, Klebsiella pneumoniae*, *Acinetobacter baumannii*, and *Aspergillus flavus, Aspergillus fumigatus, Aspergillus niger,* etc. have been identified. Furthermore, *Pseudomonas aeruginosa* and *E. coli* are the most common multidrug-resistant (MDR) isolated pathogens associated with hospital-acquired superinfections [33,35,36].

It is necessary to carry out quick diagnostic tools and systematic screening of COVID-19 patients who had also been diagnosed with bacterial coinfection to choose the appropriate antimicrobial therapy to reduce the severe complications and limit the spread of drug-resistant bacteria. [33,34].

## 6. Maternal, Fetal and Neonatal Microbiota

The human body provides nutrition and a suitable living environment for the microbiota, which plays an important role in the immune, metabolic, endocrine systems, etc. Dynamic and complex changes in the microbiota can be attributed to the host’s physiological and pathological states, especially immunity, inflammation, metabolic environment, etc. In recent years, many systemic and multifactorial diseases, such as diabetes and inflammatory bowel disease, have been shown to be associated with abnormal microbial communities, a term dysbiosis [36,37,38].

A healthy pregnancy is a complex physiological process, accompanied by coordinated responses from multiple organ systems, including weight gain, decreased blood pressure and immune tolerance, and increased hormone levels (progesterone, estradiol, prolactin), as well as IL-6, fibrinogen, and clotting factors VII-X [37].

These physiological changes during pregnancy can cause adaptive changes in the composition of the maternal microbial flora. For example, increasing the estrogen level increases the production of glycogen in the vagina, which subsequently affects the dynamics of the vaginal microbiota. Progesterone could alter the intestinal microbial structure during pregnancy, such as increasing the abundance of *Bifidobacterium*. On the other hand, immune alterations also impact microbial composition. For example, growing IL-15 levels could reduce the abundance of butyrate-producing bacteria. Abnormal changes in the maternal microbiota have been associated with gestational complications, such as preeclampsia (PE), gestational diabetes mellitus (GDM), and preterm birth (PTB) [37].

Vertical transmission of microorganisms from the maternal body, including the skin, vagina, breast milk, and intestine, to the fetus, contributes to developing the infant’s gut microbiota, with rising evidence suggesting in utero influence. It has been assumed that the uterus is a sterile environment; however, some studies demonstrate a unique placental microbiome [38].

On the other hand, maternal gut strains are more enduring in the baby’s gut and are better ecologically adapted [38,39]. During pregnancy, the bacterial diversity in the vagina is low but stable. It could be a source of another series of microorganisms that reach the placenta, the amniotic fluid, and the fetus. It is well-known that meconium, placenta, and amniotic fluid are not sterile and harbor both cultured and uncultured microbes [40].

Therefore, it is highly probable that these microorganisms from the vagina influence neonatal immunity programming. For example, the immunological differences between neonates born vaginal versus neonates born from cesarean section could result from the influence of the vaginal microbiota. Offspring born from vaginal delivery and whose mothers were vaginally colonized with *Lactobacillus* during pregnancy had increased ratios of CD45RO+ cells and lower IL-12 levels in cord blood, indicating that lactobacilli in the vagina impacted fetal immune development. It is a possibility that this influence could be attributable to bacterial metabolites, ascending organisms, or the dependence of the vaginal microbiome on the gut microbiota [38,39].

Concerning SARS-CoV-2, its effect on pregnant women and its impact on implantation, fetal growth and development, and labor and neonatal health alter the regular physiological changes of pregnancy, significantly affecting the immune system, the respiratory system, cardiovascular function, and coagulation. These can have positive or negative consequences on the progression of COVID-19 disease. Despite inadequate evidence of vertical transmission, SARS-CoV-2 has been detected in the placenta, umbilical cord blood, and amniotic fluid, although it has not been associated with any maternal or neonatal characteristics. The low proportion of placental viral load does not appear to produce an inflammatory response, suggesting that the placenta is not a preferential target and inclusive could be a protective barrier [2,41,42,43].

There is proof to support the theory that the fetal microbiota can grow in the uterus through the placental barrier or by the ingestion of amniotic fluid, influencing the development of the fetal immune system. However, there is conflicting evidence about amniotic fluid sterility, as some authors have found microbiota and others mention that it has a sterility state [40,44]. Results from human and animal models have demonstrated that meconium becomes colonized with bacteria during pregnancy. The fetal intestines may be in contact with commensal microbes and the products of their metabolism in ingested amniotic fluid, promoting early immune development, though further studies should be carried out to confirm this situation [38].

As mentioned earlier, microbiota colonization is indispensable for metabolic and hormonal homeostasis and immune maturation during normal pregnancy. In healthy pregnancies, the colonization of the microbiota of the intrauterine cavity originates exclusively in the ascending pathway through the urogenital tract (urinary, cervical, and vaginal) and the hematogenous route through the placenta after translocation from the digestive tract (oral and intestinal). The oral and intestinal microbiota stability is affected by extrinsic factors, especially diet, which influence the dynamics of the cervicovaginal microbiota [45,46].

Preterm newborns obtain a distinct microbiota compared to full-term newborns in the earliest weeks of life due to diverse factors, including birth mode, lengthy hospital stays, and exposure to several antibiotics. Ghazanfari et al., 2020 reported that in preterm infants, *Enterococcus* spp., *Enterobacter* spp., and *Klebsiella* spp. are mainly present, and commensals bacteria such as *Lactobacillus* spp. and *Bifidobacterium* spp. are mostly diminished in the gut of premature babies compared to term infants [47].

## 7. Antibiotics and COVID-19

Pregnant women are often excluded from clinical trials due to the possible risk of toxicity or side effects, resulting in a lack of knowledge about the use of medications and treatments during pregnancy [48].

Antibiotics have reduced the burden of many infectious diseases since their discovery. However, their incorrect use has contributed to the appearance of significant adverse effects that affect health and the economy. On the other hand, the general increase in administering these drugs has also increased their misuse and excessive use during pregnancy. In this regard, Cantarutti et al., 2021; Arco-Torres et al., 2021 and Qu et al., 2021 reported that approximately 20–40% of these were associated with the consumption of antibiotics [32,49,50]. Furthermore, its unacceptable use during pregnancy has been associated with severe adverse events, including allergic reactions, cerebral palsy, functional impairment, cardiac arrhythmias, and maternal/neonatal death [49].

Concerning the use of antibiotics for COVID-19 treatment, several trials have been performed to address the effectiveness of these drugs in the clinical outcomes. Most of these studies have evaluated the effects of azithromycin in different stages of the disease. Popp et al., 2021, in their meta-analysis, studied antibiotics’ efficacy/safety in out- and inpatients randomized controlled trials (RCTs) in which antibiotics were compared to non-treatment, placebo, standard care, or another treatment intervention with proven efficacy for COVID-19. They reported that after 28 days of azithromycin treatment no reduction in death risks in hospitalized patients. Moreover, they informed no benefits in moderate/severe disease inpatients and low-certainty benefits when treated with azithromycin. Non-evidence for other antibiotics from RCTs is available for COVID-19 treatment [51].

Furthermore, reports indicate that 3.5% of pregnant patients who presented COVID-19, in turn, had bacterial coinfection at hospital admission, while post-hospitalization infections occurred in up to 15%. Although current evidence on bacterial infections in COVID-19 is limited, it supports the restrictive use of these drugs, especially at admission. To limit the use of antibiotics, every effort should be made to obtain blood culture and sputum samples to reinforce their use [52].

## 8. Bacterial Resistance

Antibiotic resistance represents a health and socio-economic crisis, recognized as a severe threat that affects the entire world. Overuse of these drugs increases the spread of multidrug-resistant bacteria, leading to difficult-to-treat resistant infections. This resistance, mainly of the acquired type, is a major clinical problem. Acquired resistance can occur by horizontal gene transfer between bacteria (community settings), vertical transmission between the mother and her offspring at birth and during lactation, or spontaneously due to exposure to antibiotics. In comparison, there have been multiple studies on horizontal gene transfer, not many studies on vertical transmission. Vertical transmission is important as the first bacterial colonization of infants impacts their health and immune programming throughout life [48].

With extended antibiotics administration in pregnancy and lactation, there is a risk that resistant bacteria will emerge, both in the maternal intestine and in breast milk, which can be transmitted to the infant. As priorly mentioned, antibiotic administration during pregnancy can reach the fetus by crossing the placental barrier, as well as through ingestion in lactation due to the high liposolubility and low molecular weight of these drugs. Until now, almost all the investigations have focused on the detrimental effects of antibiotics for the mother or fetus; however, there is less evidence relating to aspects of resistance to these. The presence of antibiotic resistance genes, such as extended-spectrum beta-lactamases [(ESBL, bla SHV, bla TEM, and bla CTX-M, bla OXA) and other types are transmitted by plasmids and are often found in transposons and integrons, facilitating their mobilization with another resistance mechanism. In the case of transposons, these contain insertion sequences, which usually have genes for resistance to antibiotics such as Tn9 (chloramphenicol), Tn10 (tetracycline), and Tn903 and Tn5 (kanamycin) [47,48].

Moreover, integrins are a family of potentially mobile genetic elements capable of integrating and mobilizing genetic cassettes that confer resistance to antibiotics, such as the sulI and qacEA1 genes (sulfonamides and quaternary ammonium and ethidium bromide components, respectively) [47].

In the case of RA bacteria present in the newborn’s intestine, the lack of exposure to these drugs could indicate the passage of these strains from the mother to the infant. Therefore, it is necessary to characterize species and strains with resistance genes, considering that microbiota established in early life shows a pivotal role throw offspring’s health, metabolism, immune development, etc., and might compromise antibiotic treatment efficacy in the individual lifespan [48].

## 9. Findings in the HRAEI Cultures

In the High Specialty Regional Hospital of Ixtapaluca, from May 2020 to August 2021, a total of 50 pregnant women with SARS-CoV-2 infection were admitted. They had an average age of 27 years (17–48 years) and two pregnancies (1–4), mean gestational age of 34 weeks (13–40.6 weeks), and were classified as mild to moderate in 42 patients and critical in 8. Critically ill patients were admitted to intensive care, and only 5 required mechanical ventilation. The pregnancy was of a single product in 47 patients and twins in 3, of which 51 newborns were obtained. The pregnancy resolution was performed by cesarean section in 38 patients, delivery in 7, curettage in 3, and 2 of them died upon admission to the hospital, and where the presence of death product was confirmed, so the pregnancy was not interrupted. Neonatal sex was 31 males and 20 females; the average birth weight was 2 616 g (830–4790 g). At birth, the gestational age was 37 weeks on average (28–41 weeks for Capurro). Only seven neonates had a positive PCR, whereas in the twin pregnancies, the PCR was negative in 2 of them, and in 1, one of the twins showed positive the other negative. There were no neonatal deaths.

In the High Specialty Regional Hospital of Ixtapaluca, according to the protocol for the care of pregnant women with COVID-19—which is based on the protocol for patients with hospital-acquired and ventilator-associated pneumonia—and before giving treatment with antibiotics, a battery of cultures is carried out at hospital admission (cervicovaginal, blood culture, urine culture, and stool culture). This is since in many of the COVID-19 cases at the beginning of the pandemic, Azithromycin and Doxycycline were used indiscriminately, which in addition to not having given positive results for the intubation and/or death of the patients, it had caused the generation of cross-resistance, as well as superinfection by resistant microorganisms (fungi, bacteria, and parasites) responsible for other infections associated with this pathology [53,54,55]. It should be emphasized that none of the patients attended at the hospital were treated with antibiotics before their admission.

The bacteriological culture was carried out on all patients (*n* = 50), 7 of which were positive, some for Gram-positive bacteria and others for Gram-negative; this represented 14% of the total of pregnant patients treated by COVID-19 (Table 2). It is important to mention that from the seven positive cases with a coinfection of SARS-CoV-2 and bacteria, all the pregnant women received the specifically sensitive antibiotic treatment for each case with an adequate clinical response to eliminate the bacteria (Table 2). The eradication of the pathogen was corroborated with a culture test 7 days after the treatment was finished.

According to the origin, the types of cultures recommended for each patient were cervicovaginal, blood culture, urine culture, and stool culture, of which two (cervicovaginal and urine culture) from the same patient were positive for the same patient bacteria (*S. agalatiae*). The Gram-positive bacteria identified were *Staphylococcus haemolyticus*, *Staphylococcus epidermidis*, *Streptococcus agalactiae*, and Gram-negative bacteria were *Escherichia coli* ESBL +, biotype 1 and 2, *Acinetobacter jahnsonii*, *Enterococcus faecium*, and *Clostridium difficile*. An *E. coli* ESBL + was isolated, which was treated with Meropemen as an alternative given in the antibiogram, as for *S. hemolyticus*, it was treated with Vancomycin, *S. epidermidis* with Clindamycin, *C. difficile* with Metronidazole, *A. johnsonii*, and *E. coli*, both biotype 1 and 2 with Meropenem and *S. agalactiae* with Ampicillin

## 10. Conclusions

Despite their recommendation for their in vitro activity, Azithromycin and Doxycycline have not shown a beneficial effect in treating SARS-CoV-2 infection since, from the beginning of the pandemic, they were used indiscriminately, giving cross-reactions.

Additionally, antibiotics in pregnant or lactating women should not be considered part of the initial treatment of SARS-CoV-2 infection since the vertical transmission of these drugs and microorganisms resistant to them from mother to child harms the development and succession of the infant’s gut microbiota. As mentioned above, the administration of these drugs, if not required, can favor the growth of microorganisms associated with diseases and hinder the response to treatment.

With this in mind, it is recommendable to perform several and continuous representative cultures with antibiogram (blood, cervicovaginal, sputum, urine, among others) for the patients’ follow-up. If bacterial infections result from the cultures of the patients, it is suggested to prescribe antibiotics in pregnant women with severe illness. It is advised for patients with secondary bacterial respiratory infection to follow the guidelines associated with antibacterial treatment for patients with hospital-acquired and ventilator-associated pneumonia. In the case of suspected or demonstrated presence of respiratory bacterial infection, the proposed treatment should last at least five days in patients with SARS-CoV-2 until the improvement in the signs, symptoms, and inflammatory markers. In addition, it is suggested to perform a post-treatment culture (after seven days) to confirm the elimination of the pathogen found.

Finally, the impact of the indiscriminate use of antibiotics during pregnancy can alter both the maternal and fetal-neonatal microbiota, generating future repercussions in the subjects. The alterations in the healthy microbiota in both, have been demonstrated to produce several diseases, including neurodevelopmental impairment, immunological, allergic and metabolic diseases, inflammatory bowel disease, and other infections in the offspring.

## Figures and Tables

**Figure 1 antibiotics-11-00252-f001:**
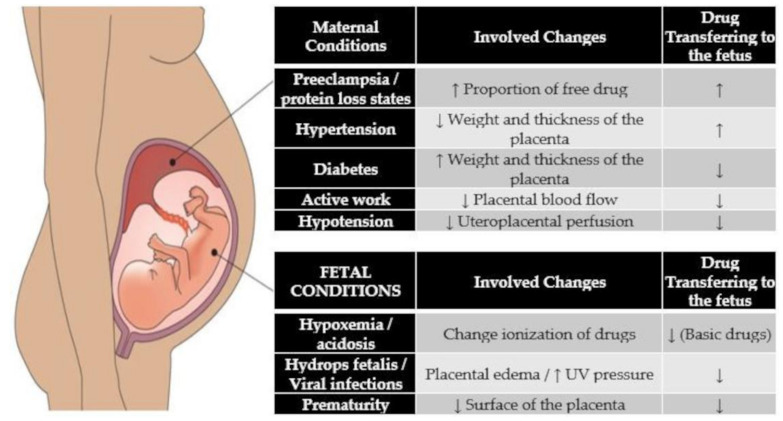
Common conditions affecting the transplacental passage of drugs, UV, umbilical vein; Da, Daltons.

**Table 1 antibiotics-11-00252-t001:** Physiological changes in response to medications during pregnancy.

Body System	Physiological Change	Effect
**Digestive System**	Slower intestinal transit, delayed gastric emptying, increased gastric pH and increased gastrointestinal blood flow	Altered drug bioavailability and delayed timing of peak levels after oral administration; although stable oral bioavailability for most drugs
	Decreased plasma albumin concentration	Increase in the drug-free fraction
	Altered CYP450 and UGT activity	Altered oral bioavailability and hepatic elimination
**Cardiovascular System**	Increased cardiac output	Increased hepatic blood flow; increase in the elimination of some drugs; affects changes in skin and muscle blood flow, which supposedly increases subcutaneous and intramuscular drug absorption; increased blood flow accelerates the rate of initiation of intravenous drugs
	Decreased epidural space due to venous engorgement	Decreases the required dose of local anesthetics
**Respiratory System**	Increased pulmonary blood flow and increased respiratory rate	Allow a higher rate of uptake of volatile anesthetics and a decrease in the time until the onset of the effect
**Endocrine System**	Increased body fat	Decreased elimination of fat-soluble drugs; increased *V*_d_ for hydrophobic drugs
**Urinary System**	Increased renal blood flow and glomerular filtration rate	Increased renal clearance
	Increased total body water and extracellular fluid	Altered disposition of the drug; increased *V*_d_ for hydrophilic drugs

UGT, uridine diphosphate glucuronosyltransferase; *V*_d_, volume of distribution.

**Table 2 antibiotics-11-00252-t002:** Bacteria isolated by culture type and antibiotic sensitivity.

#	Cervicovaginal	Blood Culture	Urine Culture	Stool Culture	Antibiotic Sensitivity	Antibiotic Resistance
**1**		*S. haemolyticus*			Rifampicin Tigecycline Vancomycin	TMP/SMX^ Doxycycline Gentamicin
**2**		*S. epidermidis*			Clindamycin Daptomycin Tigecycline	Gentamicin Oxacillin Erythromycin
				Antigen GDH* for *C. difficile*	Metronidazole Vancomycin	
		*A. johnsonii*			Imipenem Meropenem Doripenem	Ceftazidime
**3**			*E. faecium*		Vancomycin Doxycycline Linezolid	Ampicillin Ciprofloxacin Levofloxacin
			*E. coli*		Meropenem Amikacina Norfloxacin	TMP/SMX^ Cefuroxime Ceftriaxone
**4**			*E. coli* (biotype1)		Meropem Ertapenem Gentamicin	TMP/SMX^ Ampicillin Norfloxacin
			*E. coli* (biotype2)		Meropem Ertapenem Gentamicin	TMP/SMX^ Ceftriaxone Cefuroxime
**5**	*S. agalactiae*		*S. agalactiae*		Benzylpenicillin Ampicillin Vancomycin	Clindamycin
**6**	*E. coli*				Meropenem Ciprofloxacin Ertapenem	
**7**			*E. coli* ESBL +		Ciprofloxacin Meropenem Ertapenem	Ceftriaxone Cefuroxime Ampicillin
	*E. coli* (Biotype1)				Meropenem Ertapenem Gentamicin	Ceftriaxone Cefuroxime Ampicillin
	*E. coli* (Biotype2)				Meropenem Ertapenem Gentamicin	TMP/SMX Ceftriaxone Ampicillin

TMP/SMX^, Trimethoprim with Sulfamethoxazole; *GDH, glutamate dehydrogenase.

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
