# Peer review of "Does the Fetus Limit Antibiotic Treatment in Pregnant Patients with COVID-19?"

_antibiotics, 2022, doi:10.3390/antibiotics11020252_

Round 1

Reviewer 1 Report

This is an interesting and relevant review in the era of COVID pandemic. The authors described the findings of antibiogram, resistant to multiple drugs in pregnancy. Therefore, the important of performing an antibiogram in empirical treatment with antibiotics in pregnant women diagnosed with SARS-CoV-2.

Introduction should include information on the incidence of COVID in pregnancy. Is it more common for pregnant women to be infected with COVID? If yes, provide the reason. Is ACE2 playing any role in the infectivity?

Line 93: This sentence is not clear and is not grammatically correct.

It is worth mentioning that the presence of these transporters fluctuates during pregnancy; what can modulate the concentration of drugs that cross the placenta; therefore, gestational age may influence active transfer.

Suggest

It is worth mentioning that the presence of these transporters fluctuates during pregnancy; hence, what can modulate the concentration of drugs that cross the placenta may varies in different stages of gestational age.

Line 119

The authors mentioned that about 80% of the pregnancy prescriptions are antibiotics. Perhaps it would be interesting to note, the common indications for antibiotics prescriptions and what type of antibiotics are prescribed.

Line 120

The estimation of 20-25% of pregnancy received antibiotics treatment is based on just one article in 2018 (Furfaro et al. Frontier microbiology). This article is 3 years old. Antibiotics in pregnancy is a rapidly evolving field. The authors need to provide a more comprehensive review on this to give this statement.

Table 2.

There are many other conditions that could lead to placenta oedema, not just hydrops fetalis. Should consider list the other causes.

Explain how precocity could results in increased surface area.

Line 163:

The subtitle for this paragraph is (5. Effects on the maternal, fetal and neonatal microbiota). However, the discussion does not (minimally?) touch fetal and neonatal parts. It should be expanded to include the how microbiota can affect the fetus and neonates and vice versa.

Also, the discussion on the maternal part was only on gut and vagina. More importantly, the amniotic fluid microbiota should also be discussed. This is one of the areas of study lately.

Subtitles 5 and 6 are very similar?

In subtitle 5, what does effects means?

Line 174.

Is there a full stop after clotting factors.? Or should not be there?

Line 191.

It should be on the other hand, and not in the other hand.

Line 219

As noted, fetal microbiota develops from ingestion of amniotic fluid. It would be good to include a discussion of amniotic fluid microbiota or is it sterile?

Could ascending pathway from urogenital tract gain access into the amniotic cavity? And form the amniotic fluid microbiota?

Line 243

The report was more than a decade ago, should provide a more recent study.

Conclusion

Line 348

Please state the complication of alter maternal fetal neonatal microbiota.

Author Response

Reviewer’s comments, author responses, and manuscript changes

We are thankful to the referees for carefully reviewing the manuscript and the opinions regarding its presentation. In what follows, the referee’s comments are in italics, the author's responses are in blue, and the changes made are highlighted.

Reviewer 1

  1. This is an interesting and relevant review in the era of COVID pandemic. The authors described the findings of antibiogram, resistant to multiple drugs in pregnancy. Therefore, the important of performing an antibiogram in empirical treatment with antibiotics in pregnant women diagnosed with SARS-CoV-2.

Response: We are thankful for the time and effort you have invested in revising our manuscript. All your suggestions have enriched our work. Please find our answers to your valuable recommendations; we hope that we have addressed all your concerns.

  1. Introduction should include information on the incidence of COVID in pregnancy. Is it more common for pregnant women to be infected with COVID? If yes, provide the reason. Is ACE2 playing any role in the infectivity?

Response: This is an excellent observation. We added information in "Section 1. Introduction" and corrected the paragraphs as follows:

According to the most recent information, it seems there is no higher risk of catching COVID-19 in pregnant women. As for SARS-CoV-2 positive pregnant women, reports indicate that about 80-90% are asymptomatic, and only 13-15% present mild symptomatology (fever or other symptoms). Apparently, pregnant women are unlikely to experience serious illness, with only 5-8% who get a severe disease and 1-3% becoming critical [7,8].

  1. 7. https://www.who.int/docs/default-source/coronaviruse/who-china-joint-mission-on-covid-19-final-report.pdf. Consultada 29 de enero de 2022.
  2. Sutton, D.; Fuchs, K.; D'Alton, M.; Goffman, D. Universal Screening for SARS-CoV-2 in Women Admitted for Delivery. The New England journal of medicine 2020, 382, 2163–2164. https://doi.org/10.1056/NEJMc2009316.

Also, in this new version, we have extended the information about the ACE2 possible role in the complications in pregnant women positive to SARS-CoV-2.

However, reports indicate that when a condition like a viral infection is at risk, this security, preterm labor, or even fetal death are a possibility [9,10]. Pregnant women with Covid-19 had an increased risk of preeclampsia, preterm birth, and other adverse effects during pregnancy [11,12].

A possible explanation for the higher risk of pregnancy complications in infected SARS-Cov-2 women can be associated with the increase in the expression of ACE2 during gestation [13,14]. ACE is the central part of the renin-angiotensin system (RAS) and a crucial mechanism of blood pressure in mammals. ACE2 regulates vascular tension modulated by RAS because it can decompose ANG I to produce ANG 1–9, its most pleiotropic element, and lyse ANG II to ANG 1–7 (vasodilatory bioactive peptide) [15-17]. Also, SARS-CoV-2 uses ACE2 to penetrate host cells [18]. Clinical and in vivo studies reported a significantly greater expression of ANG 1–7 and ACE2 in the kidney, placenta, and uterus during pregnancy than non-pregnant species [13,14]. Nevertheless, in pregnancy, the association between ACE2 upregulation and SARS-CoV-2 needs further study [19].

  1. Silasi, M.; Cardenas, I.; Kwon, J.Y.; Racicot, K.; Aldo, P.; Mor, G. Viral Infections During Pregnancy. American Journal of Reproductive Immunology 2015, 73, 199–213. doi:10.1111/aji.12355.
  2. Alberca, R.W.; Pereira, N.Z.; Oliveira, L.M.D.S.; Gozzi-Silva, S.C.; Sato, M.N. Pregnancy, Viral Infection, and COVID-19. Front. Immunol. 2020, 11, 1672. doi: 10.3389/fimmu.2020.016 72.
  3. Wei, S. Q.; Bilodeau-Bertrand, M.; Liu, S.; Auger, N. The impact of COVID-19 on pregnancy outcomes: a systematic review and meta-analysis. Canadian Medical Association Journal 2021, 193, E540–E548.
  4. Shu Qin Wei, Marianne Bilodeau-Bertrand, Shiliang Liu and Nathalie Auge. Canadian Medical Association journal2021, 193, E540-E548; DOI: https://doi.org/10.1503/cmaj.202604.
  5. Joyner, J.; Neves, L.A.; Granger, J.P.; Alexander, B.T.; Merrill DC, Chappell MC, Ferrario CM, Davis WP, Brosnihan KB. Temporal-spatial expression of ANG-(1-7) and angiotensin-converting enzyme 2 in the kidney of normal and hypertensive pregnant rats. Am J Physiol Regul Integr Comp Physiol. 2007;293(1):R169–R177.
  6. Levy A, Yagil Y, Bursztyn M, Barkalifa R, Scharf S, Yagil C. ACE2 expression and activity are enhanced during pregnancy. Am J Physiol Regul Integr Comp Physiol. 2008;295(6):R1953–R1961. doi: 10.1152/ajpregu.90592.2008.
  7. Donoghue M, Hsieh F, Baronas E, Godbout K, Gosselin M, Stagliano N, Donovan M, Woolf B, Robison K, Jeyaseelan R, Breitbart RE, Acton S. A novel angiotensin-converting enzyme-related carboxypeptidase (ACE2) converts angiotensin I to angiotensin 1-9. Circ Res. 2000;87(5):E1–E9. doi: 10.1161/01.RES.87.5.e1.
  8. Zisman LS, Keller RS, Weaver B, Lin Q, Speth R, Bristow MR, Canver CC. Increased angiotensin-(1-7)-forming activity in failing

human heart ventricles: evidence for upregulation of the angiotensin-converting enzyme homologue      ACE2. CIRCULATION. 2003;108(14):1707–1712.

  1. Averill DB, Ishiyama Y, Chappell MC, Ferrario CM. Cardiac angiotensin-(1-7) in ischemic cardiomyopathy. CIRCULATION. 2003;108(17):2141–2146.
  2. Zhao, X., Jiang, Y., Zhao, Y., Xi, H., Liu, C., Qu, F., & Feng, X. (2020). Analysis of the susceptibility to COVID-19 in pregnancy and recommendations on potential drug screening. European journal of clinical microbiology & infectious diseases: official publication of the European Society of Clinical Microbiology, 39(7), 1209–1220. https://doi.org/10.1007/s10096-020-03897-6

  1. Line 93: This sentence is not clear and is not grammatically correct.

It is worth mentioning that the presence of these transporters fluctuates during pregnancy; what can modulate the concentration of drugs that cross the placenta; therefore, gestational age may influence active transfer.

Response: Thank you for your observation. We have attended the kind suggestion in this new version: section 3. Passage through the placenta and rephrased the sentence as follows:

It is worth mentioning that the presence of these transporters fluctuates during pregnancy; hence, what can modulate the concentration of drugs that cross the placenta may vary in different stages of gestational age.

  1. Line 119:

The authors mentioned that about 80% of pregnancy prescriptions are antibiotics. Perhaps it would be interesting to note the common indications for antibiotics prescriptions and what type of antibiotics are prescribed.

  1. Line 120:

The estimation that 20-25% of pregnancies received antibiotic treatment is based on just one article in 2018 (Furfaro et al. Frontier microbiology). This article is 3 years old. Antibiotics in pregnancy is a rapidly evolving field. The authors need to provide a more comprehensive review on this to give this statement.

Response: We thank the reviewer for this observation. In this new version, we have extended the information about prescriptions of antibiotics during pregnancy.

Reports indicate that approximately 80% of prescriptions during pregnancy are antibiotics, and it is assumed that between 20-39% of women will receive antibiotics during pregnancy [30-32]. During pregnancy, different antibacterial regimens are used in clinical practice, including cephalosporins, ẞ-lactams, macrolides, fluoroquinolones, glycopeptides, and lincosamides [30-32]. However, its ability to cross the placenta and reach the fetal compartments is variable, derived from conditions that affect the placenta, such as maternal diseases and gestational age (Table 2) [28].

  1. Furfaro, L.L.; Chang, B.J.; Payne, M.S. Applications for Bacteriophage Therapy during Pregnancy and the Perinatal Period. Front. Microbiol. 2018, 8, 2660. doi: 10.3389/fmicb.2017.02660.
  2. Yu, P. A., Tran, E. L., Parker, C. M., Kim, H.-J., Yee, E. L., Smith, P. W., … Meaney-Delman, D. (2020). Safety of Antimicrobials During Pregnancy: A Systematic Review of Antimicrobials Considered for Treatment and Postexposure Prophylaxis of Plague. Clinical Infectious Diseases, 70(Supplement_1), S37–S50. doi:10.1093/cid/ciz1231.
  3. Cantarutti, A.; Rea, F.; Franchi, M.; Beccalli, B.; Locatelli, A.; Corrao, G. Use of Antibiotic Treatment in Pregnancy and the Risk of Several Neonatal Outcomes: A Population-Based Study. Int. J. Environ. Res. Public Health 2021, 18, 12621. https://doi.org/10.3390/ ijerph182312621.

  1. Table 2.

There are many other conditions that could lead to placenta oedema, not just hydrops fetalis. Should consider list the other causes.

Explain how precocity could results in increased surface area.

Response: We thank the reviewer for this observation. In this new version, we have changed Table 2 for an image explaining the information suggested by Reviewer 3. It is worth mentioning that we added Viral infections as an inducer of placenta edema. No other cause was added because they are not related to Covid-19 disease. Regarding precocity, it was an error in translation. We have corrected precocity to prematurity.

Figure 1. Common conditions affecting the transplacental passage of drugs [28].

  1. Line 163:

The subtitle for this paragraph is (5. Effects on the maternal, fetal and neonatal microbiota). However, the discussion does not (minimally?) touch fetal and neonatal parts. It should be expanded to include how microbiota can affect the fetus and neonates and vice versa.

Subtitles 5 and 6 are very similar?

In subtitle 5, what does effects means?

Response: Thank you for the opportunity to clarify this point. We added information in section 6. Maternal, fetal, and neonatal microbiota and corrected the paragraphs as follows:

  1. Maternal, fetal and neonatal microbiota

The human body provides nutrition and a suitable living environment for the microbiota, which plays an essential role in the immune, metabolic, endocrine systems, etc. Dynamic and complex changes in the microbiota can be attributed to the host's physiological and pathological states, especially immunity, inflammation, metabolic environment, etc. In recent years, many systemic and multifactorial diseases, such as diabetes and inflammatory bowel disease, have shown to be associated with abnormal microbial communities, a term dysbiosis [36,37,38].

A healthy pregnancy is a complex physiological process, accompanied by coordinated responses from multiple organ systems, including weight gain, decreased blood pressure and immune tolerance, and increased hormone levels (progesterone, estradiol, prolactin), as well as IL-6, fibrinogen, and clotting factors VII-X [37].

These physiological changes during pregnancy can cause adaptive changes in the composition of the maternal microbial flora. For example, increasing the estrogen level increases glycogen production in the vagina, which subsequently affects the dynamics of the vaginal microbiota. Progesterone could alter the intestinal microbial structure during pregnancy, such as increasing the abundance of Bifidobacterium. On the other hand, immune alterations also impact microbial composition. For example, growing IL-15 levels could reduce the quantity of butyrate-producing bacteria. Abnormal changes in the maternal microbiota have been associated with gestational complications, such as preeclampsia (PE), gestational diabetes mellitus (GDM), and preterm birth (PTB) [37].

  1. Mohapatra, R.K.; Dhama, K.; Mishra, S.; Sarangi, A.K.; Kandi, V.; Tiwari, R.; Pintilie, L. The microbiota-related coinfections in COVID-19 patients: a real challenge. Beni Suef Univ J Basic Appl Sci. 2021, 10, 47. doi: 10.1186/s43088-021-00134-7.
  2. Zou, Y.; Qi, H.; Yin, N. Adaptations and alterations of maternal microbiota: From physiology to pathology. Med. Microecol. 2021, 9, 100045. doi.org/10.1016/j.medmic.2021.100045.
  3. Nyangahu, D.D.; Jaspan, H.B. Influence of maternal microbiota during pregnancy on infant immunity. Clin Exp Immunol. 2019, 198, 47-56. doi: 10.1111/cei.13331.

  1. Line 174:

Is there a full stop after clotting factors.? Or should not be there?

Response: Thank you for the opportunity to clarify this point. We have corrected the paragraph as follows:

A healthy pregnancy is a complex physiological process, accompanied by coordinated responses from multiple organ systems, including weight gain, decreased blood pressure and immune tolerance, and increased hormone levels (progesterone, estradiol, prolactin), as well as IL-6, fibrinogen, and clotting factors VII-X [37].

  1. Line 191:

It should be on the other hand, and not in the other hand.

Response: We thank the reviewer for this observation. We have corrected the misspelling.

  1. Line 219:

As noted, fetal microbiota develops from ingestion of amniotic fluid. It would be good to include a discussion of amniotic fluid microbiota or is it sterile?

Response: Thank you for the opportunity to clarify this point. The sterility of the amniotic fluid has been discussed, which is still somewhat controversial at this time.

There is proof to support the theory that the fetal microbiota can grow in the uterus through the placental barrier or by the ingestion of amniotic fluid, influencing the development of the fetal immune system. However, there is conflicting evidence about amniotic fluid sterility, as some authors have found microbiota and others mention that it has a sterility state [40,44]). Results from human and animal models have demonstrated that meconium becomes colonized with bacteria during pregnancy. The fetal intestines may contact commensal microbes and the products of their metabolism in ingested amniotic fluid, promoting early immune development. However, further studies should be carried out to confirm this situation [38].

  1. Singh A, Mittal M. Neonatal microbiome - a brief review. J Matern Fetal Neonatal Med. 2020 Nov;33(22):3841-3848. doi: 10.1080/14767058.2019.1583738. Epub 2019 March 5. PMID: 30835585
  2. Lim, E.S., Rodriguez, C. & Holtz, L.R. Amniotic fluid from healthy term pregnancies does not harbor a detectable microbial community. Microbiome 6, 87 (2018). https://doi.org/10.1186/s40168-018-0475-7

Continuation of question 10: Could ascending pathway from urogenital tract gain access into the amniotic cavity? And form the amniotic fluid microbiota?

Response: This route is not contemplated since there is a barrier to avoid contamination of the amniotic cavity, which is the mucous plug. In the absence of this mucous plug, no bacteria could enter the amniotic cavity.

  1. Line 243:

The report was more than a decade ago, should provide a more recent study.

Response: We thank the reviewer for this observation. We added the most up-to-date information in section 8. Antibiotics and COVID-19 and corrected the paragraphs as follows:

In this regard, Cantarutti et al., 2021; Arco-Torres et al., 2021 and Qu et al., 2021 reported that approximately 20- 40% of these were associated with the consumption of antibiotics [32,49,50].

  1. Cantarutti, A.; Rea, F.; Franchi, M.; Beccalli, B.; Locatelli, A.; Corrao, G. Use of Antibiotic Treatment in Pregnancy and the Risk of Several Neonatal Outcomes: A Population-Based Study. Int. J. Environ. Res. Public Health 2021, 18, 12621. https://doi.org/10.3390/ ijerph182312621.
  2. Arco-Torres, A.; Cortés-Martín, J.; Tovar-Gálvez, M.I.; Montiel-Troya, M.; Riquelme-Gallego, B.; Rodríguez-Blanque, R. Pharmacological Treatments against COVID-19 in Pregnant Women. J. Clin. Med. 2021, 10, 4896. doi: 10.3390/jcm10214896.
  3. Wenhao Qu1 2, Linsheng Liu 1, Liyan Miao 1Exposure to antibiotics during pregnancy alters offspring outcomes. Expert Opin Drug Metab Toxicol. 2021 Oct;17(10):1165-1174.

  1. Conclusion:

Line 348

Please state the complication of alter maternal fetal neonatal microbiota.

Response: Thank you for the opportunity to clarify this point. We added information in section 10. Conclusions and corrected the paragraphs as follows:

The alterations in the healthy microbiota in both, have demonstrated to produce several diseases, including neurodevelopmental impairment, immunological, allergic and metabolic diseases, inflammatory bowel disease, and other infections in the offspring.

Reviewer 2 Report

Dear Authors,

the article is very useful providing the summary of major impacts of the use of antibiotics during pregnancy. Administration of appropriate antibiotic should be based on the results of obtained samples always weighing the risks and benefits for the mother and child. Therefore the fetus limits the use of antibiotics in pregnancy in general, not only in Covid infection. I find the information about the most freaquent bacteria which are superimposed on Covid infection demanding specific treatment very useful. Also, it would be useful to elaborate the metabolism of drugs in damaged placenta by mentioned pathologic conditions during pregnancy.

From obtained samples You proved that most often used antibiotics were unadequate due to    bacterial resistency for which the treatment was inefficient. It is not clear weather the samples were obtained during admission to the Hospital or after. Also, were the patients treated by the protocol antibiotics before admission? The administration of adequate drugs had nothing to do with the fetus. The problem was the  wrong treatment. Maybe the title should better reflect the point of the study or in Conclusion section should be edited the answer to the question from the title.  

Author Response

Reviewer 2

  1. The article is very useful providing the summary of major impacts of the use of antibiotics during pregnancy. Administration of appropriate antibiotic should be based on the results of obtained samples always weighing the risks and benefits for the mother and child. Therefore, the fetus limits the use of antibiotics in pregnancy in general, not only in Covid infection. I find the information about the most frequent bacteria which are superimposed on Covid infection demanding specific treatment very useful. Also, it would be useful to elaborate the metabolism of drugs in damaged placenta by mentioned pathologic conditions during pregnancy.

Response: We are thankful for the time and effort you have invested in revising our manuscript. All your suggestions have enriched our work. Please find our answers to your valuable recommendations; we hope that we have addressed all your concerns.

  1. From obtained samples you proved that antibiotics most often were unadequate due to bacterial resistency for which the treatment was inefficient. It is not clear whether the samples were obtained during admission to the Hospital or after. Also, were the patients treated by the protocol antibiotics before admission?

Response: Thank you for the opportunity to clarify this point. We added information in section 10. Findings in the HRAEI cultures and corrected the paragraphs as follows:

In the High Specialty Regional Hospital of Ixtapaluca, according to the protocol for the care of pregnant women with COVID-19; which is based in the protocol for patients with hospital-acquired and ventilator-associated pneumonia; and before giving treatment with antibiotics, a battery of cultures is carried out at the hospital admission (cervicovaginal, blood culture, urine culture, and stool culture).…..It should be emphasized that none of the patients attended at the hospital were treated with antibiotics before their admission.

The bacteriological culture was carried out on all patients (n = 50), 7 of which were positive, some for gram-positive bacteria and others for gram-negative; this represented 14% of the total of pregnant patients treated by COVID-19 (Table 2). It is important to mention that from the seven positive cases with a coinfection of SARS-Cov-2 and bacteria, all the pregnant women received the specifically sensitive antibiotic treatment for each case with an adequate clinical response to eliminate the bacteria (Table 2). The eradication of the pathogen was corroborated with a culture test 7 days after the treatment was finished.

Continuation question 1. The administration of adequate drugs had nothing to do with the fetus. The problem was the wrong treatment. Maybe the title should better reflect the point of the study, or in Conclusion section should be edited the answer to the question from the title. 

Response: We thank the reviewer for this observation. We understand the reviewer's concern about the subject address in the present article. Some clinicians are always afraid to prescribe several drugs, including antibiotics, during pregnancy and lactation and even more in women with preexistent pathologies such as Covid-19. With this in mind, we rewrote section 11. Conclusions and corrected the paragraphs as follows:

Additionally, antibiotics in pregnant or lactating women should not be considered part of the initial treatment of SARS-COV2 infection. The vertical transmission of these drugs and microorganisms resistant to them from mother to child harms the development and succession of the infant's gut microbiota. As mentioned above, the administration of these drugs, if not required, can favor the growth of microorganisms associated with diseases and hinder the response to treatment.

With this in mind, it is recommended to perform several continuous representative cultures with antibiogram (blood, cervicovaginal, sputum, urine, among others) for the patients' follow-up. If bacterial infections result from the cultures of the patients, it is suggested to prescribe antibiotics in pregnant women with severe illness. It is advised for patients with secondary bacterial respiratory infection to follow the guidelines associated with antibacterial treatment for patients with hospital-acquired and ventilator-associated pneumonia. In the case of suspected or demonstrated respiratory bacterial infection, the proposed treatment should last at least five days in patients with SARS-COV2 until the improvement of the signs, symptoms, and inflammatory markers. In addition, it is suggested to perform a post-treatment culture (after seven days) to confirm the elimination of the pathogen found.

Reviewer 3 Report

1.   Type of article

The type of article is very confusing. Overall, it looked like a review article on the need of antibiotics treatment in pregnant women, however, in the abstract, it sounded like an original article looking at 50 cases of pregnant women diagnosed with COVID-19 with research findings. Suggest to make appropriate changes to the abstract in order to suit the type of article better.

2.   The authors stated briefly that pregnancy is a state of immune tolerance. Maybe can expand this fact in more details as in how these pregnant ladies are more susceptible to viral infection in introduction section.

3.   What is the current status of the use of antibiotics in COVID-19 patients? Can include more studies and current recommendations in “Antibiotics and COVID-19” section.

4.  Table 2 appears too wordy. Suggest to convert Table 2 in a form of figure.

5.   It is interesting to learn that numerous bacteria types were cultured from 50 pregnant women with SARS-CoV-2 infection.

  1. Were these women treated with antibiotics recommended? What were their responses/ outcomes?
  2. How these results can be useful? What suggestions/recommendations could be derived from there?

6.   The authors concluded that antibiotics had proven not useful and should not be used liberally without the proof of bacterial infection from culture results in the first place. Following that, what is the future prospect and research recommendations for the use of antibiotics in pregnant women infected by SARS-CoV-2? Discuss that.

Author Response

Reviewer 3

  1. Type of article

The type of article is very confusing. Overall, it looked like a review article on the need of antibiotics treatment in pregnant women, however, in the abstract, it sounded like an original article looking at 50 cases of pregnant women diagnosed with COVID-19 with research findings. Suggest to make appropriate changes to the abstract in order to suit the type of article better.

Response: We thank the reviewer for this observation. We understand the reviewer's concern about the subject address in the present article. Some clinicians are always afraid to prescribe several drugs, including antibiotics, during pregnancy and lactation and even more in women with preexistent pathologies such as Covid-19. With this in mind, we have reviewed important aspects of drug administration and implications during pregnancy and the reason to use or not antibiotics during pregnancy in patients diagnosed with SARS-CoV-2. We have rewritten the Abstract and corrected the paragraphs as follows:

Abstract: During pregnancy, there is a state of immune tolerance that predisposes them to viral infection, causing maternal-fetal vulnerability to the adverse effects of COVID-19. Bacterial coinfections significantly increase the mortality rate for COVID-19. However, it is known that all drugs, including antibiotics, will enter the fetal circulation in a variable degree despite the role of the placenta as a protective barrier and can cause teratogenesis or other malformations depending on the timing of exposure to the drug. Also, it is important to consider the impact of the indiscriminate use of antibiotics during pregnancy can alter both the maternal and fetal-neonatal microbiota, generating future repercussions in both. The literature for treating bacterial coinfections in pregnant women with COVID-19 is reviewed in the present study. In turn, we present the findings in 50 pregnant women hospitalized diagnosed with SARS-Cov-2 without previous treatment with antibiotics, bacteriological culture of sample types was performed. Seven pregnant women had coinfection with Staphylococcus haemolyticus, Staphylococcus epidermidis, Streptococcus agalactiae, Escherichia coli ESBL +, biotype 1 and 2, Acinetobacter jahnsonii, Enterococcus faecium, and Clostridium difficile. When performing the antibiogram, resistance to multiple drugs was found, such as macrolides, aminoglycosides, sulfa, dihydrofolate reductase inhibitors, beta-lactams, etc. The purpose of this study was to generate more scientific evidence on the better use of antibiotics in these patients. Because of this, it is important to perform an antibiogram to prevent abuse of empirical antibiotic treatment with antibiotics in pregnant women diagnosed with SARS-CoV-2.

  1. The authors stated briefly that pregnancy is a state of immune tolerance. Maybe can expand this fact in more details as in how these pregnant ladies are more susceptible to viral infection in introduction section.

Response: Thank you for the opportunity to clarify this point. We added information in section 1. Introduction and corrected the paragraphs as follows:

As for pregnant women, there is a state of immune tolerance during pregnancy that predisposes them to viral infection [4], causing vulnerability in both patients and newborns to the adverse effects of COVID-19 [2,5]. The recent viral outbreaks like influenza, Ebola, or the present pandemics of COVID-19 have shown that pregnant women are exposed to a worse outcome than the general population. During pregnancy, the immune system is transformed because the maternal immune system is influenced by alloantigens expressed by the placenta and the fetus. Among the changes that have been identified, there is an increased complement activity, a lower B cell function, a decrement of plasmatic pro-inflammatory cytokines (like IL-2 and interferon-gamma), and greater concentrations of anti-inflammatory cytokines like IL-4 and IL-10 have been reported [6].

  1. Abu-raya B, Michalski C, Sadarangani M and Lavoie PM(2020) Maternal Immunological Adaptation During Normal Pregnancy. Front. Immunol. 11:2627. doi:10.3389/fimmu.2020.575197.

  1. What is the current status of the use of antibiotics in COVID-19 patients? Can include more studies and current recommendations in “Antibiotics and COVID-19” section.

Response: Thank you for your kind suggestion. We have attended the kind suggestion in this new version: section 8. Antibiotics and COVID-19 and rephrased the paragraphs as follows:

Antibiotics have reduced the burden of many infectious diseases since their discovery. However, their incorrect use has contributed to the appearance of significant adverse effects that affect health and the economy. On the other hand, the general increase in administering these drugs has also increased their misuse and excessive use during pregnancy. In this regard, Cantarutti et al., 2021; Arco-Torres et al., 2021 and Qu et al., 2021 reported that approximately 20- 40% of these were associated with the consumption of antibiotics [32,49,50]. Furthermore, its unacceptable use during pregnancy has been associated with severe adverse events, including allergic reactions, cerebral palsy, functional impairment, cardiac arrhythmias, and maternal/neonatal death [49].

Concerning the use of antibiotics for the COVID-19 treatment, several trials have been performed to address the effectiveness of these drugs in the clinical outcomes. Most of these studies have evaluated the effects of azithromycin in different stages of the disease. Popp et al., 2021 studied in their meta-analysis the antibiotics efficacy/safety in out- and inpatients randomized controlled trials (RCTs) in which antibiotics were compared to non-treatment, placebo, standard care, or another treatment intervention with proven efficacy for COVID-19. They reported that after 28 days of azithromycin treatment no reduction in death risks in hospitalized patients. Also, they informed no benefits in moderate/severe disease inpatients and low-certainty benefits when treated with azithromycin. Non-evidence for other antibiotics from RCTs is available for COVID-19 treatment [51].

  1. Arco-Torres, A.; Cortés-Martín, J.; Tovar-Gálvez, M.I.; Montiel-Troya, M.; Riquelme-Gallego, B.; Rodríguez-Blanque, R. Pharmacological Treatments against COVID-19 in Pregnant Women. J. Clin. Med. 2021, 10, 4896. doi: 10.3390/jcm10214896.
  2. Wenhao Qu, Linsheng Liu, Liyan Miao. Exposure to antibiotics during pregnancy alters offspring outcomes. Expert Opin Drug Metab Toxicol. 2021 Oct;17(10):1165-1174.
  3. Popp M, Stegemann M, Riemer M, Metzendorf MI, Romero CS, Mikolajewska A, Kranke P, Meybohm P, Skoetz N, Weibel S. Antibiotics for the treatment of COVID-19. Cochrane Database Syst Rev. 2021 Oct 22;10(10):CD015025. doi: 10.1002/14651858.CD015025. PMID: 34679203; PMCID: PMC8536098

  1. Table 2 appears too wordy. Suggest to convert Table 2 in a form of figure.

Response: We thank the reviewer for this observation. We have changed Table 2 in this new version for an image explaining the information.

Figure 1. Common conditions affecting the transplacental passage of drugs [28].

  1. It is interesting to learn that numerous bacteria types were cultured from 50 pregnant women with SARS-CoV-2 infection.

  1. Were these women treated with antibiotics recommended? What were their responses/ outcomes?

Response: This is an excellent observation. We added information in section 10. Findings in the HRAEI cultures and corrected the paragraphs as follows:

In the High Specialty Regional Hospital of Ixtapaluca, according to the protocol for the care of pregnant women with COVID-19; which is based in the protocol for patients with hospital-acquired and ventilator-associated pneumonia; and before giving treatment with antibiotics, a battery of cultures is carried out at the hospital admission (cervicovaginal, blood culture, urine culture, and stool culture).…..It should be emphasized that none of the patients attended at the hospital were treated with antibiotics before their admission.

The bacteriological culture was carried out on all patients (n = 50), 7 positive, some for gram-positive bacteria and others for gram-negative; this represented 14% of the total of pregnant patients treated by COVID-19 (Table 2). It is important to mention that from the seven positive cases with a coinfection of SARS-Cov-2 and bacteria, all the pregnant women received the specifically sensitive antibiotic treatment for each case with an adequate clinical response to eliminate the bacteria (Table 2). The eradication of the pathogen was corroborated with a culture test 7 days after the treatment was finished.

  1. How these results can be useful? What suggestions/recommendations could be derived from there?

Response: Thank you for your kind suggestion. It is an excellent observation. We added information in section 11. Conclusions and corrected the paragraph as follows:

Additionally, antibiotics in pregnant or lactating women should not be considered part of the initial treatment of SARS-COV2 infection. The vertical transmission of these drugs and microorganisms resistant to them from mother to child harms the development and succession of the infant's gut microbiota. As mentioned above, the administration of these drugs, if not required, can favor the growth of microorganisms associated with diseases and hinder the response to treatment.

  1. The authors concluded that antibiotics had proven not useful and should not be used liberally without the proof of bacterial infection from culture results in the first place. Following that, what is the future prospect and research recommendations for the use of antibiotics in pregnant women infected by SARS-CoV-2? Discuss that.

Response: Thank you for your observation. We included recommendations in section 11. Conclusions and corrected the paragraph as follows:

With this in mind, it is recommended to perform several continuous representative cultures with antibiogram (blood, cervicovaginal, sputum, urine, among others) for the patients' follow-up. If bacterial infections result from the cultures of the patients, it is suggested to prescribe antibiotics in pregnant women with severe illness. It is advised for patients with secondary bacterial respiratory infection to follow the guidelines associated with antibacterial treatment for patients with hospital-acquired and ventilator-associated pneumonia. In the case of suspected or demonstrated respiratory bacterial infection, the proposed treatment should last at least five days in patients with SARS-COV2 until the improvement of the signs, symptoms, and inflammatory markers. In addition, it is suggested to perform a post-treatment culture (after seven days) to confirm the elimination of the pathogen found.
